# A Comprehensive Evaluation of Associations Between Routinely Collected Staging Information and The Response to (Chemo)Radiotherapy in Rectal Cancer

**DOI:** 10.3390/cancers13010016

**Published:** 2020-12-22

**Authors:** Klara Hammarström, Israa Imam, Artur Mezheyeuski, Joakim Ekström, Tobias Sjöblom, Bengt Glimelius

**Affiliations:** Department of Immunology, Genetics and Pathology, Uppsala University, SE-751 85 Uppsala, Sweden; israa.imam@igp.uu.se (I.I.); artur.mezheyeuski@igp.uu.se (A.M.); joakim.ekstrom@igp.uu.se (J.E.); tobias.sjoblom@igp.uu.se (T.S.); bengt.glimelius@igp.uu.se (B.G.)

**Keywords:** Rectal cancer, radiotherapy, chemoradiotherapy, response prediction, population-based, comprehensive

## Abstract

**Simple Summary:**

Rectal cancer patients are often treated with radiotherapy, either alone or combined with chemotherapy, prior to surgery to enable radical surgery on a non-resectable tumor or to lower the recurrence risk. For some patients, the tumor disappears completely after preoperative treatment, while others experience little or no benefit. Accurate prediction of therapy response before treatment is of great importance for a personalized treatment approach and intentional organ preservation. We performed a comprehensive evaluation of the predictive capacity of all routinely collected staging information at diagnosis in a population-based, completely staged patient material of 383 patients representing a real-life clinical situation. Size or stage of the rectal tumor were independent predictors of excellent response irrespective of preoperative treatment, with small/early-stage tumors being significantly more likely to reach a complete response. Levels of the tumor marker carcinoembryonic antigen (CEA) above upper normal limit halved the chance of response.

**Abstract:**

Radiotherapy (RT) or chemoradiotherapy (CRT) are frequently used in rectal cancer, sometimes resulting in complete tumor remission (CR). The predictive capacity of all clinical factors, laboratory values and magnetic resonance imaging parameters performed in routine staging was evaluated to understand what determines an excellent response to RT/CRT. A population-based cohort of 383 patients treated with short-course RT (5 × 5 Gy in one week, scRT), CRT, or scRT with chemotherapy (scRT+CT) and having either had a delay to surgery or been entered into a watch-and-wait program were included. Complete staging according to guidelines was performed and associations between investigated variables and CR rates were analyzed in univariate and multivariate analyses. In total, 17% achieved pathological or clinical CR, more often after scRT+CT and CRT than after scRT (27%, 18% and 8%, respectively, *p* < 0.001). Factors independently associated with CR included clinical tumor stage, small tumor size (<3 cm), tumor level, and low CEA-value (<3.8 μg/L). Size or stage of the rectal tumor were associated with excellent response in all therapy groups, with small or early stage tumors being significantly more likely to reach CR (*p* = 0.01 (scRT), *p* = 0.01 (CRT) and *p* = 0.02 (scRT+CT). Elevated level of carcinoembryonic antigen (CEA) halved the chance of response. Extramural vascular invasion (EMVI) and mucinous character may indicate less response to RT alone.

## 1. Introduction

Colorectal cancer (CRC) is the third most common cancer type, with a global incidence of ~1.85 million cases annually. Approximately one third of these have rectal cancer, which are often defined as tumors located up to 15 cm from the anal verge. Due to improved surgical techniques and the addition of preoperative radiotherapy (RT) or chemoradiotherapy (CRT), treatment outcomes have improved with much fewer local recurrences and slightly better survival [1]. RT/CRT is usually added to allow surgery primarily in the most advanced rectal cancers, or to lower unacceptably high local recurrences rates in slightly less advanced cancers [2]. In some instances, the RT/CRT has resulted in complete disappearance of the tumor (complete clinical response, cCR), and surgery has been postponed, a strategy known as organ preservation or watch-and-wait (W&W) [3]. At surgery, a complete pathological response (pCR) is seen in 10–20% of patients [4,5], indicating that the potential for organ preservation had been present. pCR is associated with a more favorable outcome [6]. 

Organ preservation can be either unintentional or intentional. If unintentional, the indications to give RT/CRT are not altered whereas if organ preservation is intentional, RT/CRT is given to patients in the hope that the tumor will respond and disappear. In the intentional approach, RT/CRT is given even if there is no clear need for it to render the tumor resectable or to decrease local recurrence rates. In the unintentional preservation strategy, the planned surgery is postponed after the routine evaluation (often 5–7 weeks after the end of the RT/CRT) if the tumor has entirely or almost entirely disappeared and a new evaluation is done about 12 weeks after the end of the treatment. If the tumor then has completely disappeared (cCR), the patient enters a W&W program [7,8]. This strategy has generally been accepted as part of routine care, since more than half of the completely responding patients will become long-term disease-free survivors without negative surgical consequences [9,10]. CRT on its own is, however, not without late negative consequences [11]. Accurate prediction of therapy response before treatment start would be of great importance for a more personalized treatment approach and especially for organ preservation. 

Numerous studies have explored what predicts a good response, or a pCR at surgery [4,12,13,14,15,16,17,18,19,20,21]. However, these studies have thus far been unsuccessful in detecting any clinically useful marker for response prediction. Finding relevant markers with a high probability of predicting either a sufficiently good/bad response is particularly important if an “intentional” strategy is applied when CRT is given without any other indication than to avoid surgery. Sufficiently powerful predictive markers may also be important when short-course RT (5 × 5 Gy in one week, scRT) is given, e.g., in patients not tolerating CRT. Although immediate surgery is traditionally recommended after scRT, delayed surgery is more adequate if organ preservation is aimed at if the likelihood of cCR is high. Clinical factors have not been sufficiently valuable to predict whether an attempt for organ preservation is meaningful or not, but a comprehensive evaluation of all factors routinely recorded in the diagnostic work-up has not been done, and further explorations are required. Similarly, studies have explored the predictive capability of specific characteristics identified with magnetic resonance imaging (MRI) [22], presently used for routine staging. Numerous molecular properties of the rectal tumors have also been explored, usually in rather small cohorts, and although many published studies claim that a particular biomarker is promising, none have yet been sufficiently effective, as stated in multiple reviews [14,15,16,23].

The purpose of the study was firstly to make a comprehensive evaluation of the predictive capacity of all clinical and pathological factors presently recommended for staging of rectal cancer prior to treatment decision [2,24,25] and secondly to explore whether the predictive capabilities are the same after CRT and scRT with a delay (with or without systemic chemotherapy during the waiting period). We then took advantage of a population-based, completely staged patient material representing a real-life clinical situation.

## 2. Results

### 2.1. Patient Characteristics and Frequency of Complete Tumor Regression

A total of 383 patients (Figure 1) received preoperative treatment as follows: (i) scRT with delayed surgery/W&W (*n* = 155), (ii) CRT with delayed surgery/W&W (*n* = 94), and (iii) preoperative scRT+CT with delayed surgery/W&W (*n* = 134). 

The durations of the delay after the last radiation fraction are shown in Table 1. Patient and tumor characteristics prior to treatment in both the total population and in the different therapy groups separately are presented in Table 1. Since more advanced tumors are directed to a more intense treatment regimen (CRT or scRT+CT) than less advanced tumors, treatment groups are not comparable. Patients that received scRT were older and had tumors in less advanced stages compared to patients receiving CRT or scRT+CT. Involved mesorectal fascia (MRF-positivity), extramural vascular invasion (mrEMVI) and mucinous features were more frequent among patients receiving CRT or scRT+CT compared to scRT. Equal proportions of large tumors were seen in the different treatment groups, reflecting that large tumor size is not a criterion for more intense treatment. Very small tumors (<3 cm) were, however, more often directed to scRT (12% compared to < 5% in CRT and scRT+CT groups, *p* = 0.048). 

In the study cohort, 65 patients (17%) had complete remission either clinically (cCR, *n* = 22) or pathologically (pCR, *n* = 43, for simplicity, all 65 are designated CR). Patients that received scRT+CT were more likely to reach CR than scRT and CRT-patients were (*p* < 0.001, Table 1).

### 2.2. Laboratory Parameters’ Associations with Complete Tumor Regression 

Low C-reactive protein (CRP), blood leucocytes (LPK), and carcinoembryonic antigen (CEA) were associated with CR rates in the scRT+CT group but not in any of the other groups, whereas anemia did not significantly associate with CR rate in any of the treatment groups in univariate analyses (Table 2). When the same laboratory parameters were investigated in the entire cohort, low CEA, CRP, and LPK were associated with CR (Table 3). Low CEA value (*p* = 0.032) remained associated with CR in multivariate analyses in the scRT+CT group (Table 4). CRP and LPK measures were missing to a large extent in some groups (Table 1) and therefore not included in the multivariate analysis.

### 2.3. Tumor and Nodal Stages’ Associations with Complete Tumor Regression

Clinical tumor (cT) stage was associated with treatment response in the scRT and CRT therapy groups, with an early stage tumor (cT1-2) being more likely to disappear completely after treatment (*p* = 0.006 and < 0.001, respectively, Appendix A). Decreasing proportions of CR were associated with a more advanced cT stage. This pattern was not seen in the scRT+CT group, but the limited number of patients (*n* = 8) with early stage tumors referred to this treatment regimen makes it difficult to draw conclusions. In the scRT+CT group, as many as 37% of patients with a tumor in the most advanced stage, cT4b, achieved CR after treatment (Appendix A). Further, tumors in stage cT4a–b were more likely to achieve CR after scRT+CT (27%) than after scRT (0%, *p* < 0.001) or CRT (9%, *p* = 0.026). 

When tumor stages were divided into two groups (cT1–3a and cT3b–4b), with cT3a referred to the cT1–2 group because of difficulties to radiologically distinguish cT2 tumors from cT3a tumors [26], cT stage remained associated with treatment response in the scRT and CRT treatment groups in both univariate (Table 2) and multivariate (Table 4) analyses. In both groups, tumor stage > cT3b was inversely correlated with CR (Table 4. scRT-group: OR 0.09 (95%CI 0.01–0.55), *p* = 0.010; CRT-group: OR 0.11 (0.02–0.59), *p* = 0.010).

Clinical nodal stage was associated with CR rate only in the group receiving CRT (*p* = 0.004, Table 2). However, few patients with cN0 were referred to CRT (*n* = 10) or scRT+CT (*n* = 6). Most patients in stage II (71%) received scRT without additional chemotherapy.

### 2.4. Size and Other MRI Characteristics’ Associations with Complete Tumor Regression 

Tumor size was associated with treatment response in all therapy groups, with very small tumors (<3 cm) more likely to achieve CR after treatment (*p* = 0.007, 0.006 and 0.023, respectively, Table 2). However, few of the tumors were < 3 cm in size (*n* = 27). The proportion of CR among large tumors (>5 cm) increased with more advanced treatment, from 5% in the scRT group to 19% if subsequent chemotherapy was administered (Table 2). When a cut-off of 5 cm was used, large tumors treated with scRT+CT had CR less often than the smaller tumors (*p* = 0.043, Appendix A). This was also seen in the entire cohort (*p* = 0.011) but not among patients receiving scRT or CRT. Results from multivariate analyses demonstrated that size correlated strongly with treatment response among patients receiving scRT+CT (Table 4), with considerably higher likelihood of CR if the tumor was < 3 cm in size (OR 0.10 for tumors > 5cm, (95% CI 0.01–0.66), *p* = 0.017). This was not seen for patients receiving scRT or CRT, but in a multivariate analysis of the entire cohort, size remained associated with CR. Size was evaluated in a receiver operating characteristic (ROC) analysis (Appendix A) in all treatment groups together. The three categories of tumor size, <3 cm, 3–5 cm and > 5 cm, are detailed and the cut-offs of 3 cm and 5 cm are shown as points on the tumor size ROC curve. The ROC points sit roughly the center of the ROC plot, with a spacing of 40–50% in both sensitivity and specificity, and hence the use of 3 cm with a sensitivity of 19% and specificity of 95% and 5 cm, sensitivity 65%, specificity 52%, as cut-offs for tumor size are appropriate from a ROC analysis point of view. Among patients receiving scRT, high tumor level was associated with tumor remission in the multivariate analysis (OR 14.53 for tumor level 11–15 cm, (95% CI 1.46-145.09), *p* = 0.023, Table 4). This correlation was not seen in univariate analysis (Table 2). 

Distance to the MRF in cT3 tumors did not correlate with treatment response in any of the therapy groups (Table 2). mrEMVI+ tumors demonstrated worse response to scRT compared to mrEMVI– tumors (*p* = 0.017), not seen in the CRT or scRT+CT treatment groups (Table 2). However, mrEMVI+ patients were more likely to be treated with chemotherapy in addition to RT (CRT: *n* = 44; scRT+CT: *n* = 70) compared to scRT only (*n* = 46). Mucinous features were not significantly associated with treatment response in any of the therapy groups. 

### 2.5. Factors Associated with Complete Tumor Regression

In the entire study cohort, treatment had a strong association to tumor response with increasing likelihood of CR with more intense treatment (OR 7.68, (95% CI 2.50–23.57) and 12.24 (4.25–35.24) for CRT and scRT+CT, respectively, with scRT as reference, Table 4). Besides treatment, size and CEA level were also independently associated with CR in a stepwise logistic regression model. All factors associated with tumor remission in each therapy group and the entire study cohort are presented in Table 4. The ability of the four logistic regression models to accurately predict CR was also evaluated through ROC analysis (Figure 2). The area under curve (AUC) value, which is the average highness of the ROC curve, were 0.745 (95% CI 0.678–0.807) for the entire study cohort and 0.846 (0.765–0.917) for the scRT group, demonstrating an appreciable level of accuracy in their predictive abilities. For the CRT and scRT+CT groups, the AUC values were 0.528 (0.320–0.731) and 0.583 (0.474–0.689), respectively, and hence no appreciable predictive abilities are apparent among those latter two groups. 

## 3. Discussion

In this study, statistically significant associations between several pretreatment tumor and patient specific factors and CR after scRT, CRT or scRT+CT were found. Partial response may be sufficient if the aim is to allow surgery in a non-resectable or difficult-to-resect tumor or to decrease unacceptably high local recurrence rates, but not sufficient if the aim is organ preservation. Most of the associations were seen whether the patients were treated with scRT alone or followed by systemic neoadjuvant chemotherapy or conventional long-course CRT, even if the number of patients in the three treatment groups then became rather limited and statistical significance not always reached. Independent importance for achieving CR was mainly seen for factors related to the tumor burden, i.e., stage or size of the tumor. Additional independent information was obtained by one laboratory value, CEA, but not in all treatment groups. 

The patient cohort used in this study is population based. It is, thus, not optimized for maximizing the chances of organ preservation but reflects the distribution seen in an unselected cohort where many patients have locally advanced tumors [27]. Since the aim was to find predictors of excellent response to (C)RT, it was appropriate to group cCR and pCR together. A cCR is basically a prerequisite for a pCR and if cCR is not seen, the chance of a durable response without regrowth if surgery is not done is virtually zero. Organ preservation was practiced if the response was excellent throughout the time period but was seldom practiced in patients without any other indication for pretreatment. Thus, very few early tumors, apparently responding to a much higher extent than locally advanced tumors, were included in the material; these patients were mainly treated with immediate surgery or scRT with surgery the following week. The proportions of cCR and pCR likely reflect the panorama you can see in an unselected population of rectal cancer. A publication by Hughes et al. [28] made us hesitant to recommend a W&W procedure in locally advanced tumors.

The cell kill effect of the three different treatments differs for obvious reasons and the results follow what could be expected. More intense treatment causes more cell death. The retrospective design of our study, where selection to different treatments was not done by random but rather according to patient and tumor characteristics (more advanced tumors received more advanced treatment provided the patient was fit enough), is, however, not optimal for comparing the efficacy of different treatments. Adding active chemotherapy for rectal cancer in the delay to surgery after CRT has in one study resulted in more responses [29] and it is likely that the same will be seen after scRT. More pCRs after CRT than after scRT was also seen in a Dutch retrospective study [30]. In that study, the results remained also after adjustment for prognostic factors. The higher CR-rates seen here after scRT+CT compared to after CRT are in line with the results of the RAPIDO randomized trial recently reported, doubling pCR rates (from 14% to 28%, *p* < 0.0001), and improving disease control [31]. In our study, the difference in CR rates between the scRT group and the two other groups were statistically significant (*p* = 0.01 vs *p* < 0.001) whereas it was not between the CRT and scRT+CT groups (*p* = 0.12). Many of the patients included in the group having scRT+CT participated in RAPIDO; the remaining patients were treated with fewer chemotherapy cycles (four instead of six as in RAPIDO) in the LARCT-US protocol (NCT03729687). Thus, it is not surprising that the CR-rates in the CRT and scRT+CT groups are virtually identical to those seen in the RAPIDO trial. The pCR-rates (including the cCRs) are in line with what have been reported previously after CRT, explored extensively, and after scRT alone [13,32,33].

Multiple studies have previously evaluated the factors used here for their capability to predict pCR-rates, but as far as we know, not all of them in one comprehensive evaluation. An important novelty in our study is that factors associated with CR were studied among patients treated with scRT+CT; this has not, to our knowledge, been done before. Most studies have also not categorized the continuous variables into clinically useful categories as we did. Our purpose was to explore whether the different parameters can be used practically to predict tumor response before treatment start, and not explore whether the tumor response varies mechanistically with, e.g., anemia. We have therefore used categorical variables in our analyses. Actually, most of the parameters we found associated with response rate have also correlated with pCR in other studies [12,19,34,35], although not universally. Many of the factors revealing significant associations with response have been reported after CRT before, but a comprehensive evaluation of all routinely available factors including high-quality MRI parameters has to our knowledge not been reported. The information about what factors may influence response after scRT and scRT/CRT+ neoadjuvant chemotherapy is very limited. Results from the ROC analyses demonstrated a high level of predictive ability of the regression models in the entire cohort and in the scRT group. These results suggest that treatment response to scRT can be predicted with reasonable accuracy using only a few clinical parameters. The logistic regression models for CRT and scRT+CT could however not accurately predict CR which indicates that other factors, such as those reflecting tumor cell properties, than the ones used for staging and treatment decision, are important for accurate treatment response prediction.

Tumor stage was the most important factor for whether CR was seen in the scRT and CRT treatment groups. In the scRT+CT group, size rather than stage was of greater importance, but very few early stage tumors were treated with this intense treatment. Tumor stage is used to stratify patients to pre-treatment or upfront surgery, and we have previously suggested that the border is not between cT1-2 and cT3-4, as has been the tradition for decades, but goes somewhere within the cT3 group [24]. In the absence of knowledge where that optimal cut-off is, we gave raw proportions for all substages (Appendix A). From our results it seems as if the cut-off concerning prediction of tumor response is between cT1-2 and cT3a both in the entire cohort (46% and 15% CR, respectively) and in each therapy group (scRT: 28% vs 12%; CRT: 100% vs 0%; scRT+CT: 50% vs 33%). The material is not large enough to draw firm conclusions from, and it is also retrospectively collected. Nonetheless, similar results were seen in other studies investigating factors associated with pCR after preoperative CRT [13,19,21] or scRT [36]. Tumors in stage cT2 also constitute most patients with CR after CRT [37]. A study investigating connection of cT3 invasion and pCR found that tumors in stage cT3a-b were more prone to achieve CR after CRT compared to cT3c-d (35% vs 9% pCR) [38]. MRI discrimination of cT2 and early cT3 tumors is challenging because of the difficulty of distinguishing fibrotic spread into the perirectal fat with tumor cells (cT3a) from fibrosis only (cT2) [26]. Because of this, we chose to group cT3a with cT2 in the multivariate analyses. However, including also cT3b as an early tumor yielded similar results. The relevance of this stage subgrouping may also differ according to tumor height; in tumors close to the anal canal, there is limited distance between the bowel wall and the fascia as opposed to high rectal tumors where the distance can be several centimeters.

Comparisons of smaller and larger tumors and their connection to CR rate using different cut-offs showed that very small tumors (<3 cm in size) were more prone to disappear completely after treatment in all groups, but there was a limited number of very small tumors in the study cohort. When the median size (5 cm) was used as cut-off, an association with CR was seen only among patients receiving scRT+CT. The results confirm the clinical impression that even tumors larger than 5 cm can respond completely, with increasing proportions CR with more intensive treatment, but that smaller tumors reach CR more often [39]. In the study by Garland et al including 294 patients, size was predictive (*p* = 0.008), but no cut-off was defined [13]. Taken together with previous studies, in which size has also been important, either 4 cm [17,40] or borderline 5 cm [18], size is likely most important for the ability to reach CR and thus being candidates for organ preservation. Since also large tumors can disappear completely, other aspects are important, and a bulky tumor cannot per se be exempted from attempts to organ preservation. 

Most patients had a normal CRP below 5 mg/L before start of therapy, and it was associated with CR in the scRT+CT group but not in the other therapy groups. Yasuda et al found a correlation between a higher CRP and less tumor regression in 73 patients that received CRT [41]. Interestingly, only 10% (7 out of 69) of the patients in our study cohort with CRP > 10 mg/L achieved a CR (Appendix A), indicating that there is a connection between high CRP and worse treatment response. 

Preoperative anemia in rectal cancer patients has a significant negative impact on OS and DFS [42]. Neoadjuvant treatment, both RT and CRT, is dependent on sufficient oxygen for cytotoxicity and hypoxia disables this [42]. Several studies have shown that anemia is related to a lower rate of pCR in patients receiving CRT, and an explanation for this is the generation of tumor hypoxia [12,34]. Joye et al and Clarke et al observed that patients with pCR had slightly higher mean Hb than those that did not achieve pCR, but the differences were not considered clinically relevant [12,34]. In our study, we could not see a correlation between anemia and therapy response in any of the therapy groups. 

An initially elevated CEA-value > 3.8 μg/L correlated with the rate of CR in all treatment groups, although a clear difference was seen only in the scRT+CT group. However, in all groups, the chance of CR was less than half if CEA was elevated than if normal. We did not explore whether the best cut-off for CEA is the upper normal level. CEA is believed to correlate with the tumor mass and that a larger tumor generally means a higher CEA-value [43]. As smaller tumors have a higher chance of pCR, this may be an explanation for a correlation between a low CEA-value and a higher rate of CR. CEA is one of the most frequently analyzed predictors. In a meta-analysis including 23 studies with almost 8500 patients, a low CEA level with cut-offs between 3.0 and 5.0 μg/L meant a doubling of the chance of reaching pCR than elevated levels [44], i.e., similar to our finding. Other studies have also found a correlation between elevated pre-treatment CEA level and pCR after CRT [19,21]. CEA is likely one of the most powerful predictors of response to preoperative RT/CRT and may be clinically relevant. 

All cT4-tumors are MRF+ per definition, and some cT3-tumors are MRF+. When we restricted the analyses to MRF-positivity within cT3 tumors only, we were not able to find any significant associations to CR rates in any treatment group, nor among all cT3 tumors. MRF-positivity is thus not an independent relevant parameter to consider when evaluating the chances to reach a CR. Tumor level has been explored in a few studies with variable results, although most have reported that the pCR rates are lower in low-lying tumors [40,45]. On the other hand, several studies similar to our study could not detect any difference [5,17,34]. In our study, we found that tumor level was a predictor of CR in the scRT-group, with high tumors being more likely to respond completely compared to low tumors.

EMVI is well known to negatively influence OS and local recurrence independent of tumor stage [46,47] and the vast majority of mrEMVI+ tumors had chemotherapy in addition to scRT. CRT can cause fibrosis of vessels in the tumor, which can be associated with survival outcomes [46]. There is limited knowledge if the presence of mrEMVI affects tumor response to neoadjuvant therapy; in a collaborative group study including 649 patients, mrEMVI+ before treatment was not predictive, however, only seen in 6% of the tumors [5]. In the present study, 43% of the tumors were mrEMVI+ but a connection between mrEMVI status and CR was not seen in the treatment groups receiving chemotherapy. In the scRT-group, none of the mrEMVI+ tumors (0 of 46) had CR whereas this was seen in 11% (11 of 99 patients) in mrEMVI− tumors (*p* = 0.017). No difference in CR between mucinous and non-mucinous tumors was seen in any of the treatment groups. In the scRT-group, no mucinous tumors reached CR (*p* = 0.067). However, in the groups receiving scRT+CT as many as 26% of the mucinous tumors had a CR (13 of 50 patients). This was not expected, as previous studies have indicated that rectal tumors with mucus production have poor response to CRT [20,21,35,48]. Too few patients are included in our study to be able to make firm conclusions, but if true, mucinous tumors could also reach pCR and should thus not be excluded from organ preservation, unless RT is given alone. 

There are both strengths and limitations in the present retrospective study. One important strength is the use of a large unselected population-based, i.e., a real-life patient cohort. This can however also be a disadvantage because of the heterogeneity. The total number of patients is comparably high, but as the study population is divided into groups based on therapy and sub-grouped according to the investigated variables, the material shrinks and is in some analyses too limited to draw firm conclusions. Another limitation is the high proportion of missing data in some laboratory parameters (>10%) which lead to exclusion from multivariate analyses. Variables with < 10% missing data were included in the multivariate analyses and this proportion of missing data could potentially introduce bias. However, complete case analyses were not performed due to a very limited patient number if patients with any missing data were excluded. Importantly, all factors considered in therapy guidelines [24] had few missing data and were included in the multivariate analyses.

## 4. Materials and Methods 

This study is based on a retrospective, population-based cohort study of all patients diagnosed with rectal cancer between 2010 and 2018 in Uppsala or Dalarna (total population 630,000 in 2015) at the time of diagnosis. Patient and tumor characteristics for 2010-2015 have been described [27]. The clinical and MRI-defined parameters that were considered are presented in Table 1. The quality of the staging procedures is at a high international level after several educational workshops [27,49]. A total of 959 non-metastatic patients were considered for study inclusion (Figure 1). Three hundred and eighty-three patients that received scRT, CRT or scRT with chemotherapy (scRT+CT) and either had a delay to surgery (*n* = 361) or entered a W&W program (*n* = 22) after achieving cCR were selected for treatment response analyses. All treatments were provided in routine care and both the RT or CRT were given as described in multiple publications describing the results of the trials run during the time [31,32,50].

### 4.1. Clinical and Pathological Staging and Treatment

All patients had a morphological diagnosis of invasive rectal adenocarcinoma. The rectum was defined as the most distal 15 cm of the bowel using a rigid rectoscope. Staging included clinical investigation, routine laboratory tests, computed tomography of the liver, abdomen and lungs (lung x-ray was permitted) and a “high-quality” MRI of the pelvis, as described [27]. The investigated factors are all routinely collected and considered in rectal cancer care and all patients were discussed in a multidisciplinary team meeting (MDT) prior to treatment decision. Staging and treatment recommendations followed the national care programs from 2008 and 2016 [27]. Briefly, early tumors were operated without any neo-adjuvant therapy, intermediate risk tumors received scRT with immediate surgery, or delayed surgery within the Stockholm III trial [51], and high-risk (locally advanced, ugly) received CRT (50 Gy in 5 weeks with a fluoropyrimidine, usually capecitabine, (825 mg/m^2^ twice daily during radiotherapy) with a delay to surgery. scRT was used as an alternative to CRT in unfit patients [52]. After termination of inclusion into the RAPIDO trial [50], where patients were randomized between CRT and scRT followed by six cycles of CAPOX, patients with high-risk tumors were treated with scRT followed by 4 cycles of CAPOX (scRT+CT) within the LARCT-US trial (NCT03729687). Following the publication of the Stockholm III trial results [51], scRT with a delay was more often used in intermediate risk tumors than immediate surgery.

Examination of the surgical specimen followed the routines described by Quirke [53]. If no tumor was seen in the routinely taken tumor pieces, multiple new pieces were taken and split sections at different levels done, to confirm a pCR. Tumor regression was retrospectively classified as pCR/cCR (complete disappearance of the tumor after surgery: ypT0N0 or complete remission on post-treatment MRI at least 12 weeks after the last treatment and remaining or at least 12 months: ycT0N0) or non-pCR (tumor still present after treatment). The laboratory analyses were performed at the hospital as part of clinical routine. The upper normal limit (UNL) was used as cut-off in analyses for all laboratory parameters except Hb, for which Hb < 110 g/L was used to define anemia. UNL for CEA is 3.8 μg/L. In patients with elevated CEA levels in this study (*n* = 179), the median value was 9 μg/L (range 3.9–1507 μg/L). UNL of CRP is 5 mg/L, in patients with elevated CRP levels (*n* = 120) the median value was 13 mg/L (range 5.1–140 mg/L). UNL of LPK is 9 × 10^9^/L, in patients with elevated levels (*n* = 83) the median LPK value was 10.5 × 10^9^/L (range 9.1–17.5 × 10^9^/L).

### 4.2. Statistical Methods 

Patients were grouped as follows: (i) scRT + delayed surgery/W&W, (ii) CRT + delayed surgery/W&W, and (iii) scRT + CT + delayed surgery/W&W. Different clinical and laboratory parameters and MRI characteristics and their connection to CR/non-CR were investigated within therapy groups and in the total cohort. Since the aim was to find clinically useful information, associations with continuous values were not explored, but rather categorical variables defined before analyses. The exceptions to this were clinical T-stage, where previously only T1-4 and not substages have been analyzed, and size, where, in the absence of much knowledge, an initial median value of 5 cm and subsequently two lower values (4 and 3 cm) were used. 

IBM SPSS Statistics version 27.0 and R version 4.0.3 were used for all statistical analyses and statistical significance was considered if the P value was < 0.05. Pearson’s Chi-square test was used for group comparisons to examine any associations between the chosen predictive variable and the extent of tumor regression. If more than 20% of cells had expected frequencies < 5, Fisher’s exact test was performed. Multivariate analyses of factors associated with CR in each therapy group and in the whole cohort were performed using stepwise binary logistic regression with backward elimination to estimate odds ratios (OR) and 95% confidence intervals (CI). For variable selection between steps, P values of 0.05 and 0.1 were used as inclusion and exclusion criteria, respectively. Investigated factors with > 10% missing data were excluded from these analyses. All variables with < 10% missing data were included. To avoid the effect of collinearity in the analyses, factors directly correlated to another factor were excluded (e.g., TNM stage). In additional multivariate analyses, only factors significantly associated with CR in the univariate analyses were entered on step 1 to increase robustness of the findings.

### 4.3. ROC Analyses

The four logistic regression models of Table 4 were evaluated using ROC analysis, see Figure 2. The fitted values of the four logistic regression models, along with CR/non-CR outcomes, were exported from IBM SPSS into an Excel spread sheet, and then read into the statistical software R (version 3.2.2). By computing sensitivity and specificity numbers for a range of cut-off values, each cut-off applied to the set of logistic regression model fitted values, ROC curves were obtained. AUC values were also computed. Tumor size (length in the bowel) was similarly evaluated using ROC analysis with calculation of AUC values (see Appendix A)

Confidence intervals (95%) for the AUC values were obtained using non-parametric bootstrap. The CRs and the non-CRs were resampled, yielding a new AUC-value, and by replicating the resampling 100,000 times 95% confidence intervals were obtained by taking the 2.5th and the 97.5th percentiles as the lower and upper endpoints, respectively, of the confidence interval. It should be noted that the thus obtained confidence intervals account for random sampling error only. For instance, sources of error such as selection bias or other systematic sources of error in the data collection are not accounted for. Because the ROC analyses were performed using the data that the logistic regression models were fitted to, these analyses do not constitute an independent validation.

## 5. Conclusions

Accurate prediction of therapy response before treatment start is of great importance for intentional organ preservation in rectal cancer patients. Size or stage of the rectal tumor were independently associated with CR in all therapy groups, with small or early stage tumors being considerably more likely to reach CR. In addition, strong correlations were seen between low CEA value and CR in the scRT+CT group and tumor level in the scRT-group. An elevated CEA more than halved the chance of an excellent response, and this may be clinically relevant for intentional organ preservation. An important finding is that the correlation of different factors to CR does not fundamentally differ between different therapy groups. Possible exceptions are mrEMVI+ and mucinous characteristics that may be connected to worse treatment response to radiation alone. This knowledge may facilitate the treatment decision. All factors with predictive capability are related to tumor burden, supporting a generally held view that early, small tumors are the best candidates for organ preservation. 

## Figures and Tables

**Figure 1 cancers-13-00016-f001:**
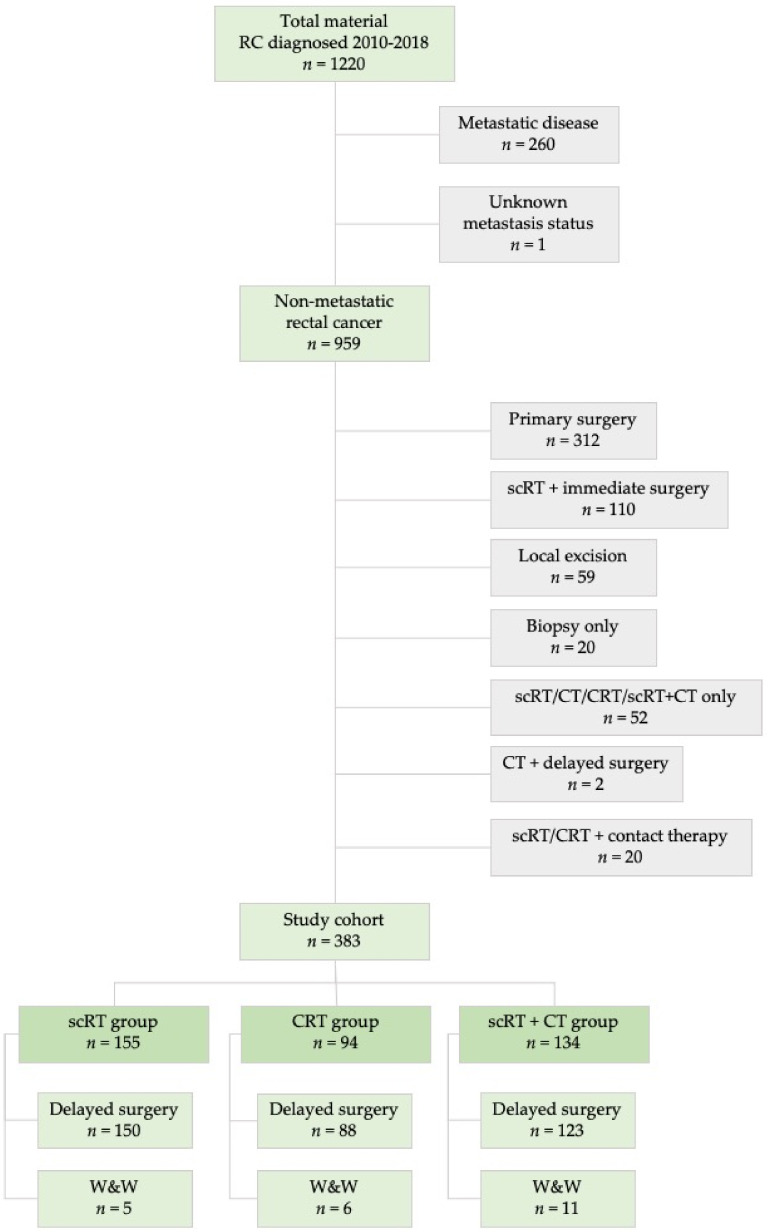
Flow chart of study cohort selection from the total cohort of 1220 rectal cancer patients diagnosed between 2010 and 2018 in two Swedish counties. Abbreviations: scRT = short-course radiotherapy; CRT = chemoradiotherapy; CT = chemotherapy; W&W = watch-and-wait.

**Figure 2 cancers-13-00016-f002:**
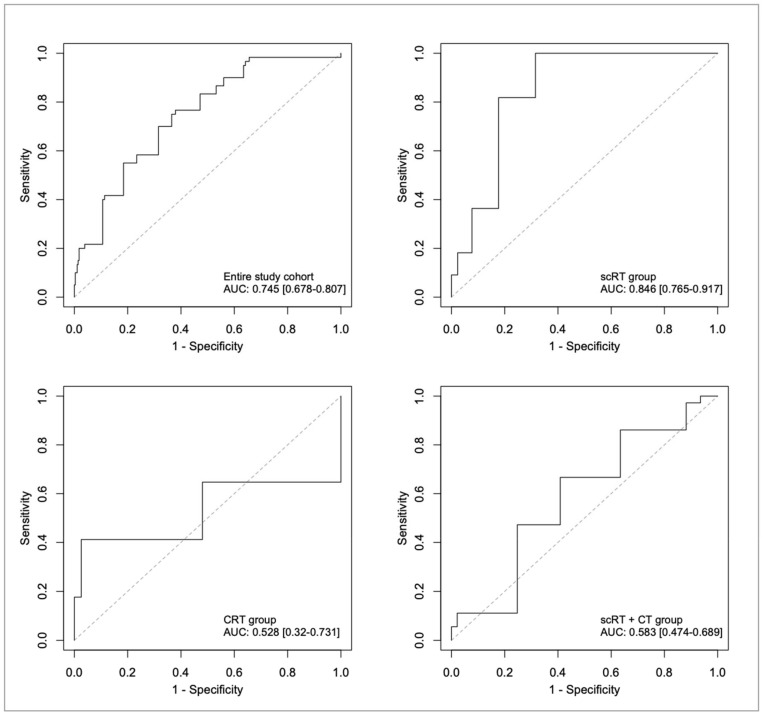
Receiver operating characteristic (ROC) curves of the four logistic regression models (solid lines) and reference ROC curves of the uninformative random classifier (dashed lines), annotated with area under curve (AUC) values and their 95% confidence intervals. (top left) Entire study cohort. (top right) scRT group. (bottom left) CRT group. (bottom right) scRT+CT group. Abbreviations: scRT = short-course radiotherapy; CRT = chemoradiotherapy; CT = chemotherapy.

**Table 1 cancers-13-00016-t001:** Clinical T and N stage, MRI characteristics and laboratory values routinely considered at diagnosis and treatment response according to treatment of non-metastatic rectal cancer patients.

*CLINICAL CHARACTERISTICS*	TOTAL POPULATION *n* = 383	scRT *n* = 155	CRT *n* = 94	scRT+CT *n* = 134	*p*-Value ^A^
AGE AT DIAGNOSIS					
Median (range) years	68 (31–91)	74 (46–91)	65 (32–80)	65 (31–81)	
<65	138 (36)	33 (21)	43 (46)	62 (46)	<0.001
65–79	194 (51)	74 (47)	49 (52)	71 (53)	
>= 80	51 (14)	48 (31)	2 (2)	1 (1)	
SEX					
Male	204 (53)	91 (59)	49 (52)	64 (48)	0.172
Female	179 (47)	64 (41)	45 (48)	70 (52)	
CLINICAL TN STAGE					
Stage I	14 (4)	9 (6)	5 (5)	0	<0.001
Stage II	42 (11)	31 (20)	5 (5)	6 (5)	
Stage III	327 (85)	115 (74)	84 (90)	128 (95)	
CLINICAL T STAGE ^B^					
cT1	4 (1)	3 (2)	1 (1)	0	<0.001
cT2	22 (6)	15 (10)	5 (5)	2 (2)	
cT3 all	208 (54)	95 (61)	44 (47)	69 (52)	
cT3a	26 (7)	17 (11)	3 (3)	6 (5)	
cT3b	66 (17)	37 (24)	9 (10)	20 (15)	
cT3c	87 (23)	30 (19)	21 (22)	36 (27)	
cT3d	24 (6)	6 (4)	11 (12)	7 (5)	
cT3 unknown	5 (1)	5 (3)	0	0	
cT4 all	148 (39)	42 (27)	44 (47)	63 (47)	
cT4a	41 (11)	10 (7)	9 (10)	22 (17)	
cT4b	107 (28)	31 (20)	35 (37)	41 (30)	
Missing cT stage ^C^	1	1	0	0	
CLINICAL N STAGE ^B^					<0.001
cN0	56 (15)	40 (26)	10 (11)	6 (5)	
cN1	143 (37)	65 (42)	24 (26)	54 (40)	
cN2	184 (48)	50 (32)	60 (64)	74 (55)	
TUMOR SIZE					
Median (range) cm	5 (1–16)	5 (1–16)	5 (1–13)	5 (1–13)	
<3 cm	27 (7)	18 (12)	3 (3)	6 (5)	0.048
3–5 cm	166 (44)	59 (39)	44 (47)	63 (48)	
>5 cm	184 (49)	74 (49)	47 (50)	63 (48)	
Missing ^C^	6	4	0	2	
TUMOR LEVEL					
0–5 cm (low)	150 (39)	66 (43)	42 (45)	42 (31)	0.083
6–10 cm (mid)	143 (37)	60 (39)	32 (34)	51 (38)	
11–15 cm (high)	90 (23)	29 (19)	20 (21)	41 (31)	
MRI-IDENTIFIED EMVI					
EMVI−	210 (57)	99 (68)	49 (53)	62 (47)	0.001
EMVI+	160 (43)	46 (32)	44 (47)	70 (53)	
Missing ^C^	13	10	1	2	
MRI-IDENTIFIED MUCINOUS FEATURES					
Mucinous−	257 (70)	110 (76)	66 (70)	81 (62)	0.029
Mucinous+	112 (30)	34 (23)	28 (30)	50 (38)	
Missing ^C^	14	11	0	3	
MRI-DETECTED MRF INVOLVEMENT IN cT3 TUMORS (*n* = 208)					
MRF negative	92 (45)	54 (59)	14 (32)	24 (35)	0.007
MRF positive/threatened	112 (55)	37 (41)	30 (68)	45 (65)	
Missing ^C^	4	4	0	0	
DELAY OF SURGERY AFTER RADIOTHERAPY ^D^					
Median (range) weeks	10 (4–36)	7 (4–33)	9 (5–28)	19 (8–36)	<0.001
<6 weeks	5 (2)	5 (4)	0	0	
6–8 weeks	95 (31)	70 (55)	25 (42)	0	
>8 weeks	206 (67)	51 (41)	34 (58)	121 (100)	
LABORATORY VALUES					
HEMOGLOBIN (Hb)					
Hb ≤ 110 g/L	50 (13)	25 (17)	10 (11)	15 (11)	0.303
Hb > 110 g/L	327 (87)	126 (83)	83 (89)	118 (89)	
Missing ^C^	6	4	1	1	
LEUCOCYTES (LPK)					
LPK ≤ 9 × 10^9^/L	247 (75)	89 (69)	73 (85)	85 (74)	0.030
LPK > 9 × 10^9^/L	83 (25)	40 (31)	13 (15)	30 (26)	
Missing ^C^	53	26	8	19	
C-REACTIVE PROTEIN (CRP)					
CRP ≤ 5 mg/L	187 (61)	65 (50)	61 (76)	61 (62)	0.001
CRP > 5 mg/L	120 (39)	64 (50)	19 (24)	37 (38)	
Missing ^C^	76	26	14	36	
CARCINOEMBRYONIC ANTIGEN (CEA)					
CEA ≤ 3.8 μg/L	176 (50)	69 (50)	40 (47)	67 (51)	0.861
CEA > 3.8 μg/L	179 (50)	69 (50)	45 (53)	65 (49)	
Missing ^C^	28	17	9	2	
TUMOR RESPONSE TO THERAPY					
COMPLETE REMISSION (CR)	65 (17)	12 (8)	17 (18)	36 (27)	<0.001
NON-COMPLETE REMISSION (NON-CR)	318 (83)	143 (92)	77 (82)	98 (73)	

Results are given numbers, *n* (%) unless indicated otherwise. ^A^ Chi-square test of independence, Fisher’s exact test was used if more than 20% of the observations had expected frequencies less than 5. ^B^ Based upon MRI in 372 patients and upon other information (mainly computed tomography (CT) and/or palpation) in 11 patients. ^C^ Patients with no information on the variable in question have been excluded in the statistical analyses. ^D^ In operated patients. Abbreviations: EMVI = extramural vascular invasion; MRF = mesorectal fascia scRT = short-course radiotherapy; CRT = chemoradiotherapy; CT = chemotherapy.

**Table 2 cancers-13-00016-t002:** Laboratory, clinical and MRI parameter’s association to treatment response in scRT, CRT and scRT+CT groups.

Clinical Characteristics		scRT			CRT			scRT+CT		
		CR	NON-CR	*p*-Value ^A^	CR	NON-CR	*p*-Value ^A^	CR	NON-CR	*p*-Value ^A^
HB	HB < 110 g/L	3 (12)	22 (88)	0.420	0			2 (13)	13 (87)	0.354
	HB > 110 g/L	9 (7)	117 (93)		17 (21)	66 (79)		34 (29)	84 (71)	
	MISSING ^B^	0	4		0	1		0	1	
CEA	CEA < 3.8 μg/L	7 (10)	62 (90)	0.165	11 (28)	29 (73)	0.103	24 (36)	43 (64)	0.025
	CEA > 3.8 μg/L	2 (3)	67 (97)		6 (13)	39 (87)		12 (19)	53 (81)	
	MISSING ^B^	3	14		0	9		0	2	
CRP	CRP < 5 mg/L	5 (8)	60 (92)	0.980	14 (23)	47 (77)	0.333	22 (36)	39 (64)	0.009
	CRP > 5 mg/L	5 (8)	59 (92)		2 (11)	17 (89)		4 (11)	33 (89)	
	MISSING ^B^	2	24		1	13		10	26	
LPK	LPK < 9 × 10^9^/L	7 (8)	82 (92)	1.000	14 (19)	59 (81)	0.450	29 (34)	56 (66)	0.035
	LPK > 9 × 10^9^/L	3 (8)	37 (92)		1 (8)	12 (92)		4 (13)	26 (87)	
	MISSING ^B^	2	24		2	6		3	16	
TUMOR STAGE ^C^	cT1-3a	7 (20)	28 (80)	0.004	6 (67)	3 (33)	0.001	3 (38)	5 (62)	0.446
	cT3b-4b	4 (4)	110 (96)		11 (13)	74 (87)		33 (26)	92 (74)	
	MISSING ^B^	1	5		0	0		0	1	
NODAL STAGE ^C^	cN0	5 (12)	35 (88)	0.328	6 (60)	4 (40)	0.004	2 (33)	4 (67)	0.584
	cN1	5 (8)	60 (92)		4 (17)	20 (83)		12 (22)	42 (78)	
	cN2	2 (4)	48 (96)		7 (12)	53 (88)		22 (30)	52 (70)	
TUMOR LEVEL	0–5 cm (LOW)	6 (9)	60 (91)	0.576	10 (24)	32 (76)	0.088	15 (36)	27 (64)	0.291
	6–10 cm (MID)	3 (5)	57 (95)		2 (6)	30 (94)		12 (24)	39 (76)	
	11–15 cm (HIGH)	3 (10)	26 (90)		5 (25)	15 (75)		9 (22)	32 (78)	
TUMOR SIZE	<3 cm	5 (28)	13 (72)	0.007	3 (100)	0	0.006	4 (67)	2 (33)	0.023
	3-5 cm	2 (3)	57 (97)		8 (18)	36 (82)		20 (32)	43 (68)	
	>5 cm	4 (5)	70 (95)		6 (13)	41 (87)		12 (19)	51 (81)	
	MISSING ^B^	1	3		0	0		0	2	
EMVI STATUS	EMVI−	11 (11)	88 (89)	0.017	8 (16)	41 (84)	0.813	18 (29)	44 (71)	0.418
	EMVI+	0	46 (100)		8 (18)	36 (82)		16 (23)	54 (77)	
	MISSING ^B^	1	9		1	0		2	0	
MUCINOUS STATUS	MUCIN−	11 (10)	99 (90)	0.067	13 (20)	53 (80)	0.770	23 (28)	58 (72)	0.765
	MUCIN+	0	34 (100)		4 (14)	24 (86)		13 (26)	37 (74)	
	MISSING ^B^	1	10		0	0		0	3	
MRF STATUS ^D^	MRF−	5 (9)	49 (91)	0.696	4 (29)	10 (71)	0.184	7 (29)	17 (71)	0.670
	MRF + (<1 MM)	2 (5)	35 (95)		3 (10)	27 (90)		11 (24)	34 (76)	
	MISSING ^B^	0	4		0	0		0	0	

Results are given numbers, *n* (%) unless indicated otherwise. Complete responders (CR) and non-complete responders (non-CR) were compared in the statistical analyses. ^A^ Chi-square test of independence, Fisher’s exact test was used if more than 20% of the observations had expected frequencies less than 5. ^B^ Patients with no information on the variable in question have been excluded in the statistical analyses ^C^ Based upon MRI in 372 patients and upon other information (mainly computed tomography (CT) and/or palpation) in 11 patients. Patients with unknown cT3 substage (*n* = 5) have been excluded from the analysis since group allocation was not possible ^D^ In cT3 tumors (*n* = 208). Abbreviations: Hb = hemoglobin; CEA = carcinoembryonic antigen; CRP = C-reactive protein; LPK = leukocytes; EMVI = extramural vascular invasion; MRF = mesorectal fascia scRT = short-course radiotherapy; CRT = chemoradiotherapy; CT = chemotherapy.

**Table 3 cancers-13-00016-t003:** Laboratory, clinical and MRI parameter’s association to treatment response in the entire study cohort.

CLINICAL CHARACTERISTICS		TOTAL MATERIAL (*N* = 383)		
		CR	Non-CR	*p*-Value ^A^
HB	Hb ≤ 110 g/L	5 (10)	45 (90)	0.146
	Hb > 110 g/L	60 (18)	267 (82)	
	MISSING ^B^	0	6	
CEA	CEA ≤ 3.8 μg/L	42 (24)	134 (76)	0.002
	CEA > 3.8 μg/L	20 (11)	159 (89)	
	MISSING ^B^	3	25	
CRP	CRP ≤ 5 mg/L	41 (22)	146 (78)	0.004
	CRP > 5 mg/L	11 (9)	109 (91)	
	MISSING ^B^	13	63	
LPK	LPK ≤ 9 × 10^9^/L	50 (20)	197 (80)	0.028
	LPK > 9 × 10^9^/L	8 (10)	75 (90)	
	MISSING ^B^	7	46	
TUMOR STAGE ^C^	cT1-3a	16 (31)	36 (69)	0.004
	cT3b-4b	48 (15)	276 (85)	
	MISSING ^B^	1	6	
NODAL STAGE	cN0	13 (23)	43 (77)	0.353
	cN1	21 (15)	122 (85)	
	cN2	31 (17)	153 (83)	
TUMOR LEVEL	0–5 cm (low)	31 (21)	119 (79)	0.116
	6–10 cm (mid)	17 (12)	126 (88)	
	11–15 cm (high)	17 (19)	73 (81)	
TUMOR SIZE	<3 cm	12 (44)	15 (56)	<0.001
	3–5 cm	30 (18)	136 (82)	
	>5 cm	22 (12)	162 (88)	
	MISSING ^B^	1	5	
EMVI STATUS	EMVI–	37 (18)	173 (82)	0.501
	EMVI+	24 (15)	136 (85)	
	MISSING ^B^	4	9	
MUCINOUS STATUS	Mucin–	47 (18)	210 (82)	0.468
	Mucin+	17 (15)	95 (85)	
	MISSING ^B^	1	13	
MRF STATUS ^D^	MRF−	16 (17)	76 (83)	0.544
	MRF + (<1 mm)	16 (14)	96 (86)	
	MISSING ^B^	0	4	
TREATMENT	scRT	12 (8)	143 (92)	<0.001
	CRT	17 (18)	77 (82)	
	scRT+CT	36 (27)	98 (73)	

Results are given numbers, *n* (%) unless indicated otherwise. Complete responders (CR) and non-complete responders (non-CR) were compared in the statistical analyses. ^A^ Chi-square test of independence ^B^ Patients with no information on the variable in question have been excluded in the statistical analyses ^C^ Based upon MRI in 372 patients and upon other information (mainly CT and/or palpation) in 11 patients. Patients with unknown cT3 substage (*n* = 5) have been excluded from the analysis since group allocation was not possible ^D^ In cT3 tumors (*n* = 208). Abbreviations: Hb = hemoglobin; CEA = carcinoembryonic antigen; CRP = C-reactive protein; LPK = leukocytes; EMVI = extramural vascular invasion; MRF = mesorectal fascia; scRT = short-course radiotherapy; CRT = chemoradiotherapy; CT = chemotherapy.

**Table 4 cancers-13-00016-t004:** Stepwise logistic regression with backward elimination model for identification of factors associated with complete tumor remission (CR) in the different treatment groups and the entire study cohort.

CLINICAL CHARACTERISTICS	scRT GROUP			
		95% CI		
PARAMETER ^A^	CR, OR ^E^	LOWER	UPPER	*p*-Value
cT STAGE				
cT1-3a	1	Ref		
cT3b-4b	0.087	0.014	0.551	0.010
TUMOR LEVEL				
0–5 cm (low)	1	Ref		
6–10 cm (mid)	2.590	0.403	16.647	0.316
11–15 cm (high)	14.534	1.456	145.088	0.023
	CRT GROUP			
		95% CI		
PARAMETER ^B^	CR, OR ^D^	LOWER	UPPER	*P*-VALUE
cT STAGE				
cT1-3a	1	Ref		
cT3b-4b	0.106	0.019	0.588	0.010
	ScRT+CT GROUP			
		95% CI		
PARAMETER ^C^	CR, OR ^D^	LOWER	UPPER	*P*-VALUE
TUMOR SIZE				
<3 cm	1	Ref		
3–5 cm	0.157	0.023	1.065	0.058
>5 cm	0.097	0.014	0.663	0.017
CEA VALUE				
CEA ≤ 3.8 μg/L	1	Ref		
CEA > 3.8 μg/L	0.392	0.166	0.924	0.032
	ENTIRE STUDY COHORT			
		95% CI		
PARAMETER ^D^	CR, OR ^D^	LOWER	UPPER	*P*-VALUE
STUDY TREATMENT				
scRT	1	Ref		
CRT	7.680	2.503	23.567	<0.001
ScRT+CT	12.235	4.248	35.239	<0.001
TUMOR SIZE				
<3 cm	1	Ref		
3–5 cm	0.209	0.063	0.690	0.010
>5 cm	0.134	0.039	0.462	0.001
CEA VALUE				
CEA ≤ 3.8μg/L	1	Ref		
CEA > 3.8 μg/L	0.506	0.264	0.966	0.039

^A^ Variables entered on step 1: Tumor level (0–5 cm (ref), 6–10 cm, 11–15 cm), cT stage (cT1-3a (ref), cT3b-4b), cN stage (cN0 (ref), cN1, cN2) Hb value (Hb > 110 (ref) vs Hb ≤ 110), EMVI status (EMVI–(ref) vs EMVI+), mucin status (mucin- (ref) vs mucin+), size (<3 cm (ref), 3–5 cm, >5 cm). 134 cases included in the analysis, 21 excluded due to missing data on any of the variables. ^B^ Variables entered on step 1: Tumor level (0–5 cm (ref), 6–10 cm, 11–15 cm), cT stage (cT1-3a (ref), cT3b-4b), cN stage (cN0 (ref), cN1, cN2) Hb value (Hb > 110 (ref) vs Hb ≤ 110), LPK value ( LPK ≤ 9 × 10^9^/L (ref) vs LPK > 9 × 10^9^/L), EMVI status (EMVI–(ref) vs EMVI+), mucin status (mucin-(ref) vs mucin+), size (<3 cm (ref), 3–5 cm, >5 cm). 85 cases included in the analyses; 9 cases excluded due to missing data. ^C^ Variables entered on step 1: Tumor level (0–5 cm (ref), 6–10 cm, 11–15 cm), cT stage (cT1-3a (ref), cT3b-4b), cN stage (cN0 (ref), cN1, cN2) Hb value (Hb > 110 (ref) vs Hb ≤ 110), CEA value (CEA ≤ 3.8 (ref) vs CEA > 3.8), EMVI status (EMVI–(ref) vs EMVI+), mucin status (mucin-(ref) vs mucin+), size (<3 cm (ref), 3–5 cm, >5 cm). 124 cases included in the analysis, 10 excluded due to missing data. ^D^ Variables entered on step 1: Tumor level (0–5 cm (ref), 6–10 cm, 11–15 cm), treatment (scRT (ref), CRT, scRT+CT), cT stage (cT1-3a (ref), cT3b-4b), cN stage (cN0 (ref), cN1, cN2) Hb value (Hb > 110 (ref) vs Hb ≤ 110), CEA value (CEA ≤ 3.8 (ref) vs CEA > 3.8), EMVI status (EMVI–(ref) vs EMVI+), mucin status (mucin-(ref) vs mucin+), size (<3 cm (ref), 3–5 cm, >5 cm). 326 cases included in the analysis, 57 excluded due to missing data. ^E^ Odds ratio obtained from multivariate analysis. Abbreviations: Hb = hemoglobin; CEA = carcinoembryonic antigen; CRP = C-reactive protein; LPK = leukocytes; EMVI = extramural vascular invasion; MRF = mesorectal fascia.

## Data Availability

The data presented in this study are available upon resonable request from the corresponding author.

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
