# Peer review of "A Comprehensive Evaluation of Associations Between Routinely Collected Staging Information and The Response to (Chemo)Radiotherapy in Rectal Cancer"

_cancers, 2020, doi:10.3390/cancers13010016_

Round 1
Reviewer 1 Report
The Authors report on a retrospective study evaluating clinical factors potentially predicting the likelihood to observe a clinical or pathological complete response in rectal cancer patients undergoing RT + CT and delayed surgery or watchful waiting strategies. The topic is of interest, sample size is sufficient, methodology is clear, results are robust and the discussion well pitched. Authors should be commended for their effort and complimented for the nice study.
Author Response
Thank you for the very positive comments.
Reviewer 2 Report
This is a very interesting study that analyzes the clinical factors involved in the incidence of complete tumor response during preoperative radiotherapy for rectal cancer from a large cohort. However, a severe problem is recognized in the acceptance of this article.
1. The authors did so much detailed statistical analysis of the data that it was difficult to understand the logic the authors wanted to state.
2. In the entire cohort, there were 65 cases of CR, but the scRT and CRT groups had only about 15 cases of CR. Therefore, a multivariate analysis of each treatment method, as done in Table 2, should not be done; we think it is better to analyze only the cohort of the scRT, scRT+CT, and CRT groups combined.
3. Why did the authors use the stepwise method in the overall multivariate analysis? The authors should select the independent variables from the previous literature.
4. In Table 4 the authors have performed the Pearson's chi-square test, but there are several tests with expected frequencies below 5; it is recommended to use Fisher's exact test.
5. The authors claim that totally 17% achieved pathological or clinical CR, more often after scRT+CT than CRT or scRT in the abstract, but this is incorrect because they did not compare the scRT+CT group with the CRT group. .If the authors want to compare the ratio of CR in the three treatments, they should perform propensity score matching and other comparisons.
6. If the authors want to obtain a cut-off value for tumor size, they would be better off using ROC analysis.
7. The authors do not make any explicit mention of radiation fields or doses in this study. Furthermore, they make no mention of the nearly 4-week difference in treatment time between the CRT and scRT groups.
Reviewer 3 Report
This is an observational study investigating the predictive performance of clinicopathological features in response to RT. The main strength of this study is the population-based nature. My comments appended below: Major comments: # I would not, at this stage, call the study design "predictive". Instead, the authors investigated associations between candidate factors and treatment response. Consider adding a C-statistic or ROC curve to model the predictive performance of the multivariable model in differentiating patients with or without good response. # How did the authors select these candidate factors in Table 1? Are they literature-based or based on availability? Please clarify. # It was not clear in the method how the authors handled missing data with no more than 10% of missingness. Did you simply do a complete case analysis? This should be acknowledged as one of the limitations as any missing data could introduce bias to the observed associations. Minor comments: # Please include exact p values and effect sizes, if possible, in the abstract for the factors identified.Author Response
Please see the attachment.

Reviewer 4 Report
The major purpose of the manuscript presented by Hammarström et al. was to provide a comprehensive evaluation of the predictive capabilities of routine clinical, pathological and laboratory markers for pathological or clinical complete tumor remission (CR) in patients with rectal cancer. More particularly, by analyzing a cohort of 383 patients treated with a short-course RT (scRT), long term chemoradiation (CRT) or scRT with chemotherapy (scRT+CT) and having either a delay to surgery or entered a watch-and-wait program, authors report on a 27% CR rate after scRT+CT, 18% after CRT and 8% following scRT. Moreover, factors associated with CR cover clinical tumor stage, small size (<3 cm), tumor level and low CEA-value while extramural vascular invasion and mucinous character may indicate less response to scRT.
In summary, the study addresses a clinically relevant and timely issue. There are no major issues regarding this manuscript as the authors used a meaningful set of analyzes on a sufficient number of patients to corroborate their findings. Overall, the data are well presented. The manuscript, however, may further benefit from inclusion of some minor revision as mentioned successive.
Minor points of criticism:
- Discussion section: Authors focused their analysis on the single endpoint tumor remission. As tumor regression grading covers a prognostic marker and individual-level surrogate for disease-free survival in patients with rectal cancer (e.g., Fokas, E. et al, J Natl Cancer Inst 2017) authors should briefly discuss their findings in the light of patient’s follow up.
- Material and methods section (line 372): Criteria of histopathological assessment of tumor regression in surgical specimen should be given in detail as not all readers will be familiar with.
Round 2
Reviewer 2 Report
The authors' responses and corrections to my comments were appropriate. My concerns in this paper have been addressed.
This is a critical study that analyzes the clinical factors involved in the incidence of complete tumor response during preoperative radiotherapy for rectal cancer from a clinical cohort. I recommend that it be accepted for publication.